# Integrating noncommunicable disease care in a public primary health care facility in North Lebanon: A qualitative study of implementation in a humanitarian crisis

Linda Abou-Abbas[1‡], Lavanya Vijayasingham[2‡], Carla Zmeter[1], Aya El Khatib[1], Grace Abou Nakad[1], Leah Anku Sanga[2], Randa S. Hamadeh[3], Pablo Perel[2], Sigiriya Aebischer Perone[4,5‡], Éimhín M. Ansbro[2‡]*

**1** International Committee of the Red Cross, Beirut, Lebanon, **2** NCD in Humanitarian Settings Special Interest Group, Centre for Global Chronic Conditions and Department of Non-communicable Disease Epidemiology, Faculty of Epidemiology and Population Health, London School of Hygiene and Tropical Medicine, London, United Kingdom, **3** Primary Healthcare and Social Health Department, Ministry of Public Health, Baabda, Lebanon, **4** International Committee of the Red Cross, Geneva, Switzerland, **5** Division of Tropical and Humanitarian Medicine, Hôpitaux Universitaires de Genève, Geneva, Switzerland

‡ These authors are joint first authors on this work. SAP and EMA are joint senior authors on this work.
* eimhin.ansbro@lshtm.ac.uk

## Abstract

Integrated care for non-communicable diseases (NCDs) may enhance care delivery, clinical impact, user-satisfaction, and access. However, evidence on integrated NCD care in humanitarian settings remains scarce. This qualitative study accompanied the first year of implementation (March 2023 to March 2024) of an integrated care programme at an International Committee of the Red Cross (ICRC)-supported, Ministry of Public Health-affiliated primary healthcare centre (PHC) in North Lebanon. A preceding theory of change (TOC) workshop on combining NCD PHC, mental health and psychosocial support (MPHSS), and rehabilitation services, identified two interdependent pathways to integrated NCD care: the multidisciplinary service pathway and the patient/family support pathway. The REAIM PRISM implementation science framework guided the study design, analysis and reporting. We interviewed service users and ICRC/PHC staff and analysed interview data alongside programme documentation using a joint thematic and constructive grounded theory approach. Overall, integrated care was feasible despite contextual challenges, including economic crisis, job losses, regional tensions, and insufficient NCD financing. A key overarching theme emerged: staff-related human values, motivation, and work ethic driving action amid personal and professional adversity. Implementation narratives described assimilating new processes, adapting to contextual challenges, and resolving process-oriented problems. In the first year, interventions under the TOC's multidisciplinary pathway—such as training, streamlined processes, and improved data systems—were prioritized, notably incorporating MHPSS and diabetic foot screening,

**Data availability statement:** Since several participants are from a vulnerable group, the data underlying the analysis are available from the senior author or the London School of Hygiene & Tropical Medicine Ethics Committee (ethics@lshtm.ac.uk) on reasonable request.

**Funding:** This study was funded by a grant from Novo Nordisk A/C to the London School of Hygiene & Tropical Medicine. LV, LS, PP (grant recipient) and EA received salary support via this grant. The specific roles of these authors are articulated in the 'author contributions' section. The funders had no role in study design, data collection and analysis, decision to publish, or preparation of the manuscript.

**Competing interests:** The authors have read the journal's policy and have the following competing interests: The LSHTM-affiliated authors received salary support via a grant from Novo Nordisk A/S (https://www.novonordisk.com/) to their institution, via the Partnering for Change Initiative, a partnership between Novo Nordisk A/S, the International Committee of the Red Cross and Red Crescent Societies, and the Danish Red Cross. This does not alter our adherence to PLOS ONE policies on sharing data and materials.

while interventions in the patient empowerment pathway were not yet implemented. PHC NCD care remained the entry point for integrated care with unidirectional referrals to collocated services. Most service users reported no change in their clinic experience; however, many positively noted improvements in programme quality, respect, and trust. Limited awareness or demand for MHPSS services was observed. Strengthening bidirectional referrals, multidisciplinary meetings, patient and caregiver empowerment, and promotion of self-care may address unmet needs and promote uptake of MHPSS and rehabilitation services.

## 1. Introduction

Noncommunicable diseases (NCDs), such as cardiovascular conditions and diabetes, account for just under 75% of global deaths [1]. Almost half occur prematurely among working-aged people (younger than 70 years old) [1]. Over 80% of premature deaths occur in low-and-middle-income countries (LMICs), which is also where most humanitarian crises and operations occur [1,2]. The number of active conflicts and acute emergencies is increasing, while humanitarian crises have become increasingly protracted [3]. Often, these contexts are beset by simultaneous, intersecting challenges, or "polycrisis", such as climate emergencies, infectious disease outbreaks, economic and financial crisis, and cultural and political tensions [4]. These dynamics mean that responses in countries affected by acute and protracted crises and health system preparedness and recovery efforts must include NCDs [2].

In Lebanon, just under 90% of deaths are attributable to NCDs, which is higher than the regional average (79%) [5]. Past shocks and political instability impeded development of the public health system and Lebanon has been highly dependent on private and non-state actors and donors to meet population health needs [6–8]. The Lebanese Ministry of Public Health (MOPH) nominally funds healthcare for around half of the population, who are not covered by private or social insurance; PHC networks provide about 25% of all outpatient care [9,10]. To strengthen public primary care, they introduced an accreditation system and integrated NCD services into PHCs within their network [11]. Rolled out nationally from 2015, the service package positions PHC general practitioners (GP) as NCD case managers, who provide continuous care and initiate specialist referral as required [11]. However, ensuring care coordination and effective referrals systems remains challenging [12–15].

Hosting over one million Syrian refugees since 2011 has imposed a heavy burden on the Lebanese health system, especially at primary care level [16]. From 2019 onwards, the financial crisis, the 2020 Beirut Port explosion, the global COVID-19 pandemic, and ongoing political instability have resulted in further health system constraints [17]. High medication and equipment costs and limited national funds have halted government NCD medication subsidies and the MOPH turned to various non-governmental organisations (NGOs) to procure PHC medicines [9]. Meanwhile, fuel, water and electricity shortages along with mounting operational costs and a brain drain of health professionals mean public facilities cannot function at full capacity

[9,16,18,19]. From October 2023, increasing cross-border tensions and armed conflict affecting South Lebanon and other parts of the country have resulted in significant humanitarian consequences. As of November 2024, the conflict has killed over 4,000 people, injured more than 16,600, and affected 1.3 million people, leading to widespread displacement and extensive infrastructure damage [20]. The number of displaced persons in Lebanon has surged from 110,000 between October 2023 and September 2024 to over 875,180 by November 2024 [21].

International, non-state and humanitarian actors, including the International Committee of the Red Cross (ICRC) have supported the MOPH in addressing the Lebanese and Syrian populations' unmet health needs [12]. Operating in Lebanon since 1967, the ICRC began to support PHC services in 2014, providing care for Syrian refugee and vulnerable local Lebanese populations, and later focussing on health system strengthening [22]. ICRC's NCD response has been guided by four principles: (1) patient-centred care, (2) continuum of care, (3) integrated approach, and (4) sustainability of response through partnership and advocacy [13]. Building on these principles and informed by research on national health needs and on a theory of change process, the ICRC piloted an integrated NCD care approach in a PHC facility in North Lebanon from 2023 [13,23].

Integrated NCD care at PHC level is advocated by the World Health Organization (WHO) as a strategy that may improve work efficiencies in resource constrained settings [23–26]. Long-term continuity, shared case management, and PHC-coordinated linkage to multi-disciplinary diagnostic, therapeutic and rehabilitation services may reduce premature mortality and disability among people living with NCDs (PLWNCDs). A recent scoping review identified only four examples of integrated mental and physical NCD services in LMIC settings, drawn from Middle East and North African contexts [27]. Such services are usually built-upon existing PHC, HIV, maternal and child health programmes and infrastructure [27]. In most cases, they involve task sharing with lower cadre health workers to efficiently manage the time of the available professional human resources for health. While feasible in principle, there is little knowledge on the effectiveness of integrated NCD care programmes in LMICs, and none, to our knowledge, from humanitarian settings [27]. Studies on integrated care from high-income countries highlight the difficulty in establishing clinical impact, but have indicated positive effects on more service-user oriented domains such as satisfaction, perceived quality of care, and ease of access to services [28–30].

Here, we explored the early implementation process of an integrated model of care for PLWNCDs in a humanitarian setting. The research, conducted in parallel to the first year of programme implementation, aimed to identify lessons learned and real-time adaptations, and support longer-term maintenance of the model.

## 2. Methods

### 2.1. Study design

We used a qualitative observational case study methodology, based on primary qualitative interview data, document review, meeting notes and observations, to explore the early implementation process of integrated care for people with target NCDs at an ICRC-supported PHC in North Lebanon. We report on the preparation and the first phase of implementation, which ran from March 2023 to March 2024.

**2.1.1. Research partnership and summary of prior phase of research.** A research partnership was established between the ICRC, Danish Red Cross (DRC), Novo Nordisk A/S, and an interdisciplinary research team from London School of Hygiene & Tropical Medicine (LSHTM), called the Partnering for Change (P4C): chronic care in humanitarian crisis initiative. Earlier research from this partnership explored the health-seeking needs and experiences of vulnerable Lebanese and Syrian people living with NCDs, residing in the North and Mount Lebanon Governorates, and those of health actors delivering NCD care [14]. Key findings included challenges in maintaining continuity of care, weak or absent referral pathways to multidisciplinary and secondary care, the clear interaction between mental health and physical NCDs, the treatment burden experienced by patients and their families, and the importance of family support in managing NCDs amidst an evolving humanitarian crisis [14].

Based on these findings, in 2020, ICRC and LSHTM collaborated on a Theory of Change (TOC) process to theorise how co-located PHC, Mental Health and Psychosocial Support (MHPSS) and Physical Rehabilitation Programme (PRP) services could be integrated into a person-centred model of care [13] (Fig 1). The programme goals or proposed long-term outcomes were: 1. Improved multidisciplinary and quality of patient centred care, 2. Strengthened integration of ICRC services with improved coordination of care and information systems, 3. Patients and family members empowered to engage proactively with health services, incorporate treatment into daily life, and make the healthiest possible choices, considering the context and available options, and 4. Improved achievement of treatment targets, reduced disability and reduced mental health symptoms.

Two key pathways were identified - a multidisciplinary service pathway (developing joint human resource (HR) processes, capacity, and a shared information/data management system) and a patient/family centred pathway [13]. A suite of outcomes, interventions, indicators and assumptions was proposed [13] (S1 File). The integrated care plan targeted management of hypertension, type-1, and type-2 diabetes at PHC level; education on coping strategies for NCDs; screening for MHPSS needs and referral for services as needed; and the management of diabetic foot through the introduction of diabetic foot screening and referral to PRP services.

## 2.2. Study context and brief programme narrative

The study was conducted at the Chabab Al Ataa Al Jazeel Association (CAJA) PHC facility in the Bireh town of the Akkar Governorate in North Lebanon, which hosted about 12% of the Syrian refugee population in Lebanon [13]. Akkar

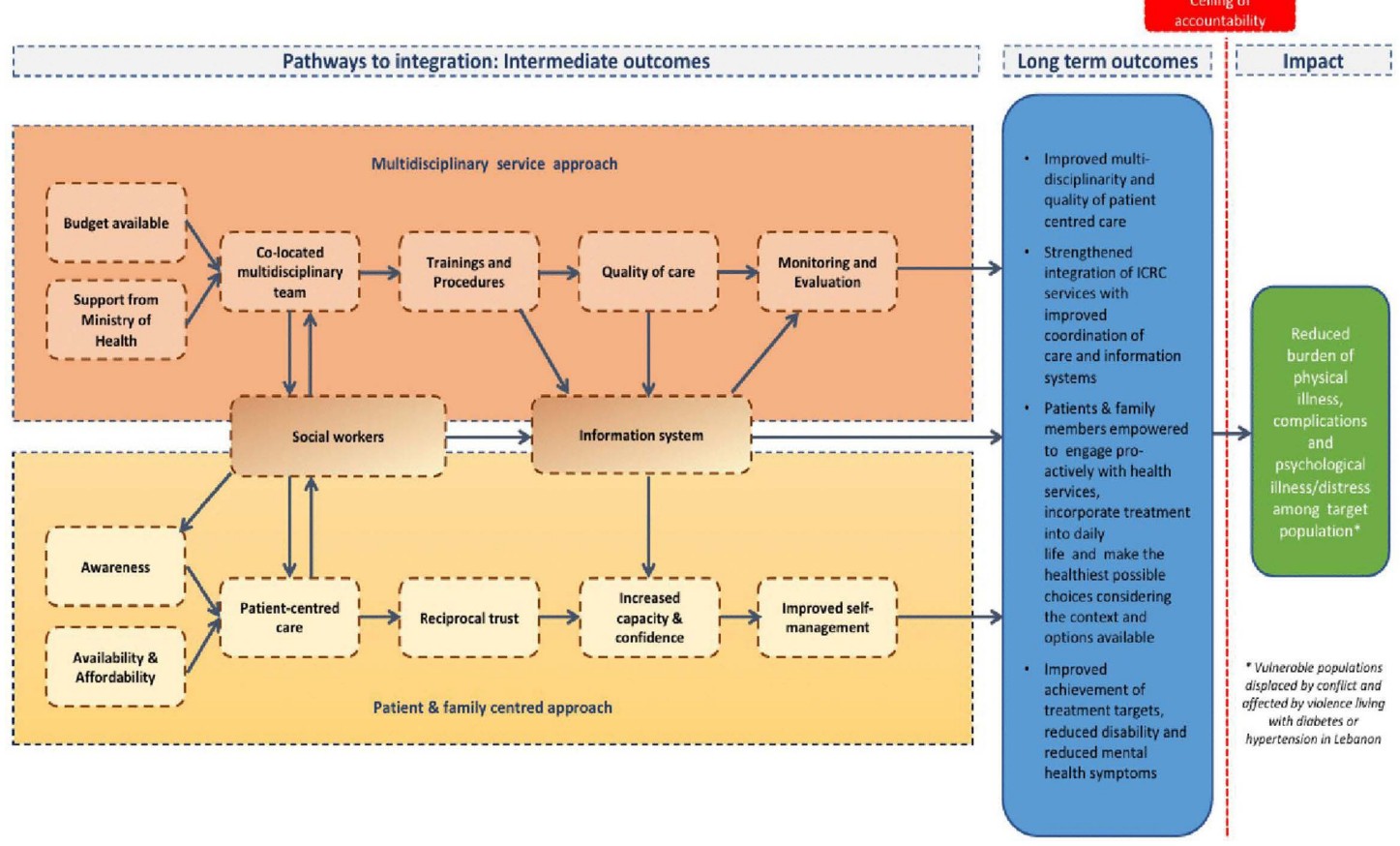

**Fig 1. Simplified theory of change map to implement integrated NCD care in CAJA [13].**

governorate holds the highest burden of NCDs in Lebanon; nearly 50% of households have a member affected by NCDs, and it is reported to have the lowest levels of access to NCD medications [31]. This governorate also faces the highest rates of food insecurity and inadequate food intake, and the highest rates of socioeconomic vulnerability (84%) among Syrian refugees [31].

In October 2019, the ICRC began financial, technical and medication procurement support to the local CAJA NGO. ICRC later identified the health facility as a suitable site for integrated NCD care. Indeed, the CAJA NGO already provided general primary care practice, and a private physiotherapy clinic was co-located within the premises, where ICRC then initiated MHPSS services. As mentioned, in 2020, ICRC and LSHTM undertook the TOC process to develop a programme theory to support the implementation of integrated primary level NCD care in CAJA. In August 2022, with ICRC support, CAJA was accredited as part of the MOPH PHC network. From 2022, as part of the P4C research partnership, following the TOC process, a research project was initiated to document the planning and first phase of implementation of integrated care in CAJA PHC.

This study was conducted within an operational humanitarian program embedded in the public PHC network, implemented under specific timing and contextual constraints, as described in the Background section.

## 2.3. Research conceptual approach and framework

The observational case study methodology involved the triangulation of multiple data sources (semi-structured interviews with different categories of participant, ICRC, MOPH and internal programme documentation; meeting notes, reflexive field notes and observations), which are described further below [32,33]. We used a joint thematic and constructivist grounded theory analytical approach, which aligns with a social constructionist paradigm, a subjectivist epistemology, a relativist ontology, and an interpretive methodology [34–36]. The joint use of these methods enabled thematic and process-oriented understanding of the implementation process of an integrated care model in humanitarian settings.

The established implementation research framework REAIM PRISM, was used to guide research conceptualisation, design, analysis and reporting [37] (Fig 2). We also evaluated the implementation process in light of the Theory of Change (intervention activities) co-developed in June 2021 [13].

## 2.4. Data sources and collection

### 2.4.1. Semi-structured interviews.
Semi-structured interviews were conducted with a total of 20 participants, including eight CAJA service users living with type 1 or type 2 diabetes and/or hypertension (Table 1), CAJA facility staff & four ICRC staff (three Lebanon-based and one Geneva headquarters-based) (Table 2) and. An interview topic guide for service users, CAJA and ICRC staff and stakeholders was co-developed by the ICRC-LSHTM team (S2 File).

All the ICRC and CAJA staff (ten women and two men) involved in the implementation of the NCD model of care were interviewed after face-to-face invitation. LSHTM researchers conducted interviews with ICRC staff (LV – PhD, female global health researcher at LSHTM), and some CAJA staff in English (LV & EA – PhD, MBBCh, female global health researcher at LSHTM and primary care physician; both experienced in qualitative research in LMIC settings) with translation support from ICRC research team (LAA – PhD, female, experienced epidemiologist and ICRC Lebanon research lead; AEK – BSc, female, ICRC Lebanon NCD officer; both experienced in qualitative operational research). The ICRC study lead interviewed remaining CAJA staff in Arabic (LAA). Two repeat interviews were conducted with ICRC Lebanon staff. From October 2023 onwards, security limitations limited further study site visits by the LSHTM team. The interviewers' backgrounds and interest in the study topic were explained to all participants. LSHTM had no prior relationship with

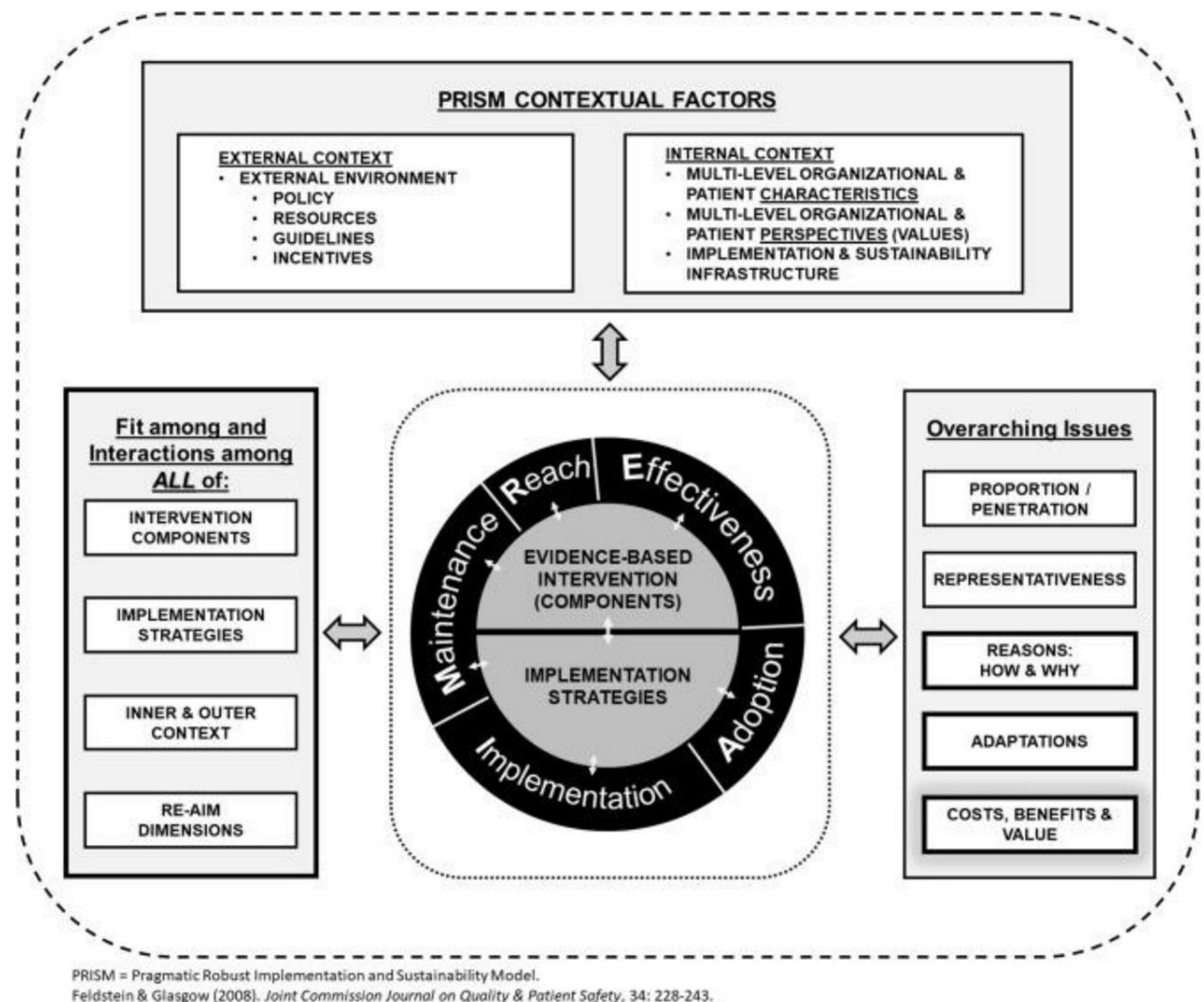

**Fig 2. REAIM-PRISM implementation research framework [37].**

stakeholder and staff participants, whereas the ICRC study team had worked with their ICRC and CAJA colleagues for several years.

Eight CAJA service users who utilised services after the integrated care programme started were selected purposefully by the ICRC and CAJA clinic team using maximum variation purposive sampling. This strategy aimed to capture a wide range of perspectives and experiences by ensuring diversity in NCD types, gender, age groups (working age 18–64 vs elderly 65 and over), nationality, mild mental health conditions, disability, and prolonged functional impairment, that may require use of prosthetics or assistive technology. Additionally, some participants were selected to contrast the decisions and experiences of uptake and refusal of the referred services. This approach ensured

**Table 1. Details of service user interview participants and interactions.**

| ID | Ages | Sex | Employ-ment | Nation-ality | Disease | Referral to | Duration attending CAJA at point of interview |
|---|---|---|---|---|---|---|---|
| 1 | 30-40 | Female | No | Syrian | DMT1 | MHPSS; attended & requested more sessions | 6-7 months |
| 2 | 30-40 | Female | No | Syrian | HTN | MHPSS, declined to receive the service | 4 months |
| 3 | 50-60 | Male | No | Lebanese | HTN; DMT2 | PRP; was referred & recommended medical shoes but was not called back & later told shoes were unavailable | 6 years |
| 4 | 50-60 | Female | Part time | Lebanese | HTN | MHPSS but declined | 4-5 years |
| 5 | 60-70 | Male | No | Lebanese | HTN; DMT2 | Psychotropic medications by GP, and individual session but did not receive the service | 2 years |
| 6 | 60-70 | Male | No | Lebanese | HTN | Referred for MHPSS but refused | 1 year |
| 7 | 50-60 | Female | No | Syrian | HTN; DMT2 | PRP; attended 1 session and declined more follow up, not open about reasons | 3 years |
| 8 | 50-60 | Male | No | Syrian | HTN; DMT2 | PRP; received & uses medical shoes-but did not receive regular PRP sessions | 3-4 years |

**MHPSS:** mental health and psychosocial support services, **PRP:** physical rehabilitation programme, **HTN:** Hypertension, **DMT1:** type 1 diabetes mellitus, **DMT2:** type 2 diabetes **mellitus.**

**Table 2. Details of CAJA PHC and ICRC staff interviews.**

| No | Role | Hiring Organization | Interview and presence of translator by | Interview dates | Interview Language | Mode |
|---|---|---|---|---|---|---|
| 1 | Admin staff | CAJA | LV and LAA | May 2023 | Arabic, with English translations | In-person |
| 2 | Medical professional | CAJA | LV and LAA | May 2023 | Arabic, with English translations | In-person |
| 3 | Medical professional | CAJA | EA and AEK | May 2023 | Arabic, with English translations | In-person |
| 4 | Medical professional | CAJA | LAA | August 2023 | Arabic | In-person |
| 5 | Admin staff | CAJA | LAA | July 2023 | Arabic | In-person |
| 6 | Medical professional | CAJA | LAA | August 2023 | Arabic | In-person |
| 7 | Medical professional | CAJA | EA and AEK | May 2023 | Arabic, with English translations | In-person |
| 8 | Leadership | CAJA | LAA | November 2023 | Arabic | In-person |
| 9 | ICRC Lebanon | ICRC | LV | July & November 2023 | English | Remote; remote |
| 10 | ICRC Lebanon | ICRC | LV | June & November 2023 | English | In-person; remote |
| 11 | ICRC Lebanon | ICRC | LV | December 2023 | English | Remote |
| 12 | ICRC HQ | ICRC | LV | November 2023 | English | Remote |

that our dataset reflected varied experiences relevant to the implementation and acceptability of the integrated care programme.

Participants were approached in the clinic waiting room and invited to participate. LAA conducted all interviews with the service users, with whom she had no prior relationship, in-person at CAJA PHC over September and October 2023 in Arabic. The interviews were between 20 and 40 minutes each. Hand written notes were taken as required. All interviews were audio recorded and transcribed and translated by independent local research consultants. We have no record of any refusals to participate in the research; transcripts were not returned to participants for their review. Data saturation was assessed during the analysis phase and was considered reached when no new themes or insights were emerging from the interviews, indicating that additional interviews were unlikely to provide substantially new information.

**2.4.2. Document review and analysis.** We used project, organizational and national policy documents; those identified by ICRC included the national MOPH NCD strategy, ICRC Lebanon's health strategy, and ICRC-LSHTM meeting notes. The documents provided insights on the general implementation process and timeline of events, decision trails and rationales, organizational context, and strategy.

**2.4.3. Memos, field notes and team reflections.** Reflections and discussion points were captured during meetings and debriefs from events, such as interviews. Researchers' and programme implementers' expertise and tacit knowledge were also captured through real-time involvement and discussions during the study period [23].

## 2.5. Data coding, analysis, and reporting

For coding and analysis, we prioritized an inductive thematic and constructivist grounded theory analysis process. This involved open and then axial coding. To generate a summative grounded theory, codes were reconstructed into coherent processes and conceptual themes, using constant comparison of data between and within data sources. Prior literature, researchers' understanding, and positionalities were also considered to influence the research lens and knowledge co-generation process. Coding and analysis were conducted manually (using MS Excel and Word for coding logs & decision trails) by LV (LSHTM). Regular team discussions were held with LSHTM and ICRC colleagues to support reflexivity, ensure analytic rigour, organic member checking among staff participants, and enhance the reliability and internal validity of interpretations. These iterative analysis meetings contributed to consensus building and refinement of emerging themes. The REAIM-PRISM framework was used to organize and frame the reporting of findings. The Consolidated Criteria for Reporting Qualitative Studies Checklist (COREQ) was used as a reporting convention [38]. Axial codes can be found in the S3 File.

## 2.6. Research team positionality and reflexivity

The ICRC-LSHTM research team is international and multi-cultural, with multidisciplinary clinical, research and local and international programme expertise, including from other humanitarian and LMIC settings. The joint lead authors are women: LAA is a local Lebanese epidemiologist (ICRC) with past work experience within the Lebanese MOPH. LV is a qualitative global health researcher (LSHTM) from a middle-income country, who has led and published several qualitative research articles on organizational processes and lived experience of chronic illness. Their reflexive field notes and broader team meeting notes are used as data sources in this analysis.

## 2.7. Patient and public involvement

People living with NCDs and their caregivers attending the CAJA PHC were involved as research participants. Other service users, the community or public were not involved in the design, conduct, analysis or reporting of the study. While member checking with service users had been planned, it could not be implemented due to time constraints and contextual challenges during the study period. However, some staff participants were engaged in multiple stages of the research and are co-authors on this paper. Their continued involvement provided an indirect opportunity for member checking, consistent with participatory research approaches that integrate validation and feedback organically throughout the research cycle. Details of the ICRC-LSHTM site visit (May 2023) was shared on the CAJA social media page in Arabic.

## 2.8. Research ethics

Research ethics board approvals were obtained from LSHTM Research Ethics Committee (LSHTM Ethics Ref: 28365), and ICRC's CORE in January 2023 (LDP_CORE 23/00008-cGB/bap). The study was conducted according to the principles expressed in the Declaration of Helsinki. Written informed consent was obtained from all study participants prior to their involvement in the research. Participants were provided with information about the study objectives, procedures,

PLOS Global Public Health

potential risks and benefits, confidentiality measures, and their right to withdraw at any time without affecting the services they received. Consent was documented through a signed consent form for both patient and staff interviews. To ensure confidentiality, all data were de-identified: participants were assigned unique codes, personal identifiers were removed from transcripts, and quotes used in the manuscript were anonymized. All data were stored securely with access restricted to the research team.

## 3. Results

Our findings are reported here in alignment with key themes from the RE-AIM PRISM conceptual framework and are then illustrated as a summative process of early implementation.

The TOC outlined two intervention pathways to achieving the programme's goals: firstly, a multidisciplinary approach that was anchored in PHC staff developing a new collaborative way of working and using a joint data system and, secondly, a patient and family centred approach. A comprehensive timeline of implementation activities and challenges is provided as supporting information (S4 File). At time of writing the ICRC were at mid-stages of implementing the proposed model of care identified through the TOC process and were preparing to hand over PHC support to the CAJA NGO by end of 2025.

### 3.1. Contextual factors: opportunities and challenges within the organizational (internal) environment

**3.1.1. Infrastructure for implementation - The integrated NCD care patient pathway at the CAJA PHC.** In March 2023, according to meeting notes and staff interviews, a streamlined patient flow for NCD patients was introduced within CAJA PHC (Fig 3).

Diabetic foot and mental health screening tools were added to the existing GP-led care, to identify NCD patients who required referral to the co-located MHPSS and PRP services, providing holistic care:

*The model consists of welcoming the patient to the center, and providing (them) with treatment in all aspects… the doctor…identifies whether the patient needs psychological or physical treatment… This model genuinely helps the patient … when they receive psychological treatment, their situation will improve, along with taking the medications. (CAJA IDI-7)*

Following check-in at reception, blood tests (implemented in accordance with MOPH guidelines) and triage, including blood pressure and capillary glucose measurement with the nurse, the team introduced the administration by the social worker of a validated Arabic translated Patient Health Questionnaire-2 (PHQ-2), a widely used two-item depression screening tool [39]. For those screening positive on PHQ2, the GP then introduced the more comprehensive PHQ-9 into the consultation [40]. He triaged the person to interventions based on their score (psychosocial interventions, psychological and psychiatric support, prescription of psychotropic medicines and psychiatric referrals). In addition to standard hypertension and diabetes management, people with diabetes now underwent a formal foot assessment, after which they were categorized into four groups, each with specific intervals for appropriate check-ups and referral to PRP for interventions, such as specialist footwear and/or insoles, physiotherapy, and education. Patients received prescribed medications from the pharmacy and ongoing education on medication adherence and healthy lifestyle practices from the social worker. Finally, the paper-based patient data were entered into the interprofessional Excel by the social worker.

**3.1.2. Organizational factors: ICRC and CAJA NGOs - human resource challenges, impact of global ICRC funding and job cuts in Lebanon.** Implementation of the integrated care model was facilitated by the ICRC Lebanon health team's strong ongoing working relationships with the MOPH, and community, leaders, and staff of the CAJA clinic that were cultivated prior to the inception of this research and intervention.

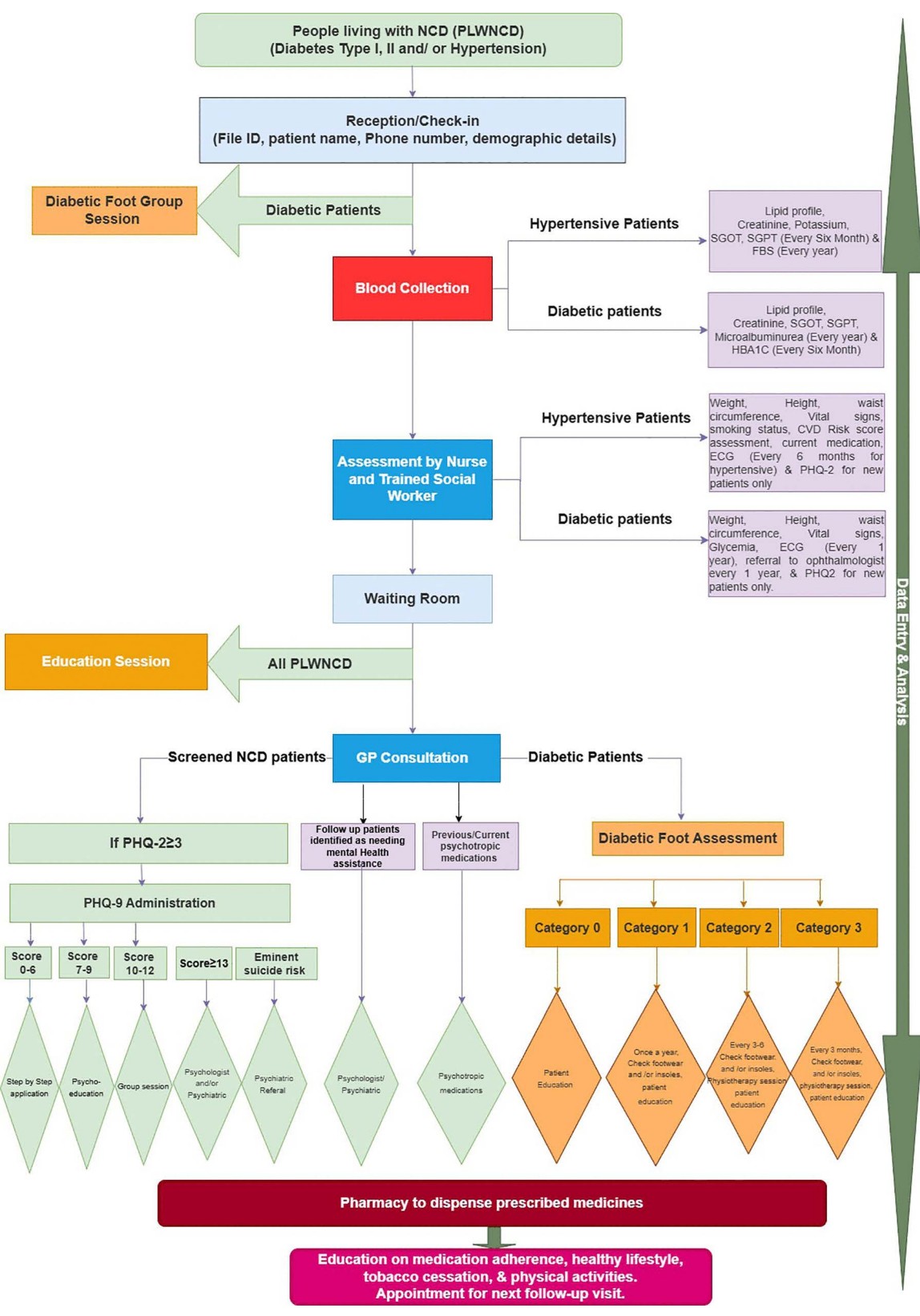

**Fig 3. Streamlined patient flow at clinic visits for integrated NCD care.**

ICRC experienced some key internal challenges during the implementation period, including high staff turnover, recruitment difficulties, reduced staffing, and reassignment of roles. This was influenced by broader trends of national health workforce emigration and movement into the private sector. In 2023, ICRC revised its institutional strategy in relation to NCDs, placing greater emphasis on acute rather than protracted crisis response. Moreover, they experienced global level financial constraints, reflecting financial challenges more broadly within the humanitarian sector. Both factors resulted in job redundancies and the reassignments of roles in Lebanon:

*This year is a completely abnormal year for the ICRC. Before, we would never have had all the HR gaps. We had the financial crisis and so that's why we lost HR and field officers for the north, and then to hire someone in an uncertain situation… It is very challenging, and you try to best allocate existing resources. (ICRC)*

The financial and recruitment constraints led to programme implementation delays and directly affected both ICRC's capacity to undertake research and the study design. They co-developed research adaptations to respond to these pressures with the LSHTM research team.

The CAJA NGO is a non-profit charitable volunteer organization established in 2011. Its mission is to support individual and community development through a diverse array of activities and initiatives. They are driven by a clear desire to serve their community, to reduce financial as well as physical suffering:

*The perspective of the dispensary for the future is alleviating people's suffering. People are tired and are facing difficult financial situations. Most people are in this situation, of course, and not to mention the existing foreign Syrian community. … All these issues make us try and search for sources that will alleviate people's suffering with our efforts, our network, and the support of the organizations. This is our hope. (CAJA)*

### 3.2. Contextual factors: Opportunities and challenges due to national (external) context

**3.2.1. Alignment with MOPH policies and programmes.** After being accredited as an MOPH PHC, the CAJA PHC was included as an MOPH pilot site for MHPSS integration into primary healthcare, based on a locally adapted version of the WHO Mental Health Gap Action Programme [13].

**3.2.2. Changing context and programme implementation delays in the age of polycrisis.** Polycrisis is defined as a "state' in which multiple, macroregional, ecologically-embedded, and inexorably interconnected systems face high – and advancing – risk across socioeconomic, political, and other dimensions" [41]. From the time of the TOC development (June 2021) to completion of year one (March 2024), the ICRC team faced several unforeseen implementation delays related to the external context. These included competing staff priorities in response to the COVID-19 pandemic and a cholera outbreak (October 2022), a worsening economy with health workforce emigration, and new waves of insecurity due to regional armed conflict. However, building on pre-existing working relationships and crisis-management experience, the ICRC were well placed to lead on responding to fast-changing realities.

### 3.3. Adoption: CAJA management and staff and patients

**3.3.1. Early acceptability and buy-in by CAJA management and staff.** The TOC process proposed several activities to facilitate the adoption of the integrated care model within ICRC and CAJA organisations, and in collaboration with the MOPH [13]. Internal ICRC processes involved securing support and budget, and the development and sharing of a programme concept note for discussion with the MOPH, facility staff and partner organizations. Interview narratives around this phase of the programme underscored the benefits of early and open communication and collaboration between all stakeholders:

*We were always in collaboration, step by step, to develop and carry out the process. They first informed us, and we agreed to provide integrated health care. We, of course, welcomed such a project. (CAJA)*

Local leaders and staff were open to new ways of operating when the likely benefits to the community were communicated and welcomed new ideas:

*Having (leadership) on your side is something that would really… I'm not really a pointing out to the power thing but having (them) convinced of the importance of this service at the PHC really helped us in terms of having the (local) staff available for this. (ICRC)*

Adoption was also facilitated by the financial incentives ICRC provided to specific CAJA staff as a strategy to incentivize and retain staff since the start of their support to the NGO. These included: the GP, triage nurse, two social workers for patient education, receptionist, ICRC focal person and the mental health social worker.

**3.3.2. Adoption of integrated NCD care by CAJA patients.** Patient narratives show that they were largely unaware of a change in processes or of a new integrated or multidisciplinary approach to care. However, individuals described undergoing diabetic foot screening, mental health assessment and receiving healthy living education. Overall, they were very positive about the quality of care they received from the CAJA team:

*I find this centre different than any other centre, in terms of the nurses, the supervisors and the doctors. They respect the patients. In other dispensaries, they do not respect the patients. (P8)*

### 3.4. Implementation of multidisciplinary approach pathway

In the first year, interventions related to the multidisciplinary approach pathway, including specific activities around staff training, implementation of diabetic foot and depression screening with related adapted clinic flow and information system changes, were prioritised (Fig 1; S3 File).

**3.4.1. Staff and human resource processes to prepare integrated service delivery.** Between May 2021 and March 2023, CAJA staff received several trainings from the ICRC and the MOPH. The ICRC provided training to CAJA staff on the management of persons living with chronic conditions, including patient education, and on multidisciplinary diabetic foot care. The MOPH provided "mhGap" mental health training for the CAJA GP and the ICRC as part of the national MHPSS programme, during which the GP was trained in brief psycho-social interventions and psychotropic medication prescribing. Specific to the integrated care programme, the social worker and nursing staff were trained in the administration of PHQ 2 & 9 questionnaires and on use of the new Excel data collection tool, described below.

**3.4.2. Multidisciplinary approach.** Initially, an informal Multi-Disciplinary Team (MDT) approach was undertaken, with ad hoc discussions between providers. For example, the nurses or social worker might flag someone with MHPSS needs to the GP. ICRC emphasised this approach in combined in-person and online training and ongoing supervision:

*Training- joint case management- this really changed their vision and their approach, and they had to work together also during the training and …we had four follow up sessions where we really asked them to also present the management of patients …. There is a lot of focus on interprofessional collaboration on teamwork, on motivational interviewing, patient education and, during the training, they had to identify … how they would better work together (ICRC).*

Formal MDT meetings were introduced later, and at the time of writing, the team planned to introduce clinical case discussions.

**3.4.3. Multidisciplinary data systems and separate reporting systems.** Prior to the inception of the integrated care model and MOPH accreditation of CAJA clinic, paper files were used to document NCD PHC patient data at each clinic visit and an Excel tool was used to monitor service delivery. The paper-based patient file was updated to reflect new changes. With support from LSHTM, the ICRC expanded their Excel tool to collect additional clinical data facilitating prospective data collection, service delivery and cohort monitoring.

The TOC process identified the need for a shared, multidisciplinary patient data collection tool to collate patient data from the three co-located services. However, the team reported that multiple ICRC and MOPH processes and requirements were in operation: 1) the ICRC-LSHTM integrated care Excel datasheet, 2) a separate ICRC MPHSS reporting system, 3) and MOPH Phenics system for central monitoring of PHC activity [42] and 4) a standalone "monthly tracking sheet" for the National Mental Health Programme. This was considered unsustainable by ICRC staff, and they hoped the pilot would serve to streamline reporting systems.

**3.4.4. Bidirectional referral pathways.** The TOC process had envisioned multiple entry points to integrated NCD care, whereby an NCD patient could be identified in any of the 3 co-located services and referred into the PHC system, where the GP served a case manager role and could refer to other collocated services. However, this was instead implemented as a single-entry point via the PHC service with unilateral referral pathways from there to PRP and MHPSS services.

While there was an increase in identification of NCD patients who could benefit from MHPSS interventions, there was an initial long waiting list for individual MHPSS sessions due to the lack of a field psychologist. Once the GP was trained in brief psychological interventions and in psychotropic prescribing and PHC staff and psychologist were in place, MHPSS sessions resumed, with the GP triaging to the appropriate intervention depending on the PHQ9 score. Group MHPSS sessions were also introduced for those with less severe symptoms to manage the backlog.

Table 3 presents the status of key qualitative indicators defined in the theory of change, as well as associated challenges, at the end of March 2024.

**Table 3. Achievement of key qualitative indicators outlined by the theory of change process and challenges at the end of March 2024.**

| Key qualitative indicators | Status | Notes |
|---|---|---|
| 1. A multi-disciplinary team (MDT) is established with social worker, nursing, medical, physio members recruited and trained. | Completed | Yes: some challenges in recruiting and retaining a psychologist were faced which influenced the start of individual MHPSS sessions. |
| 2. Clinical consultation (general practitioner), laboratory, counselling/health education (research nurse/social worker), mental health (psychologist), physiotherapy services (physiotherapist) are co-located. | Completed | Yes; on-site laboratory considered too expensive, and decision was made to outsource services. |
| 3. Bi-directional referral pathways are in place between sites of services, and exist at national and local levels, by ICRC, MOPH, working groups under UN umbrella and other health actors. | Partly | Unidirectional referral pathways to PRP and MHPSS established, with PHC as the entry point to care. External referrals to psychiatrist (10 mins away) in another humanitarian organization supported facility (International Rescue Committee) |
| 4. Means of sharing information about patients between sites and professionals of MDT is established. | Completed | Yes- though this data spreadsheet is not used for communication or shared decision making between MDT. It mainly serves as a longitudinal outcomes database of attending service users. Additionally, MOPH requires separate data sharing on their electronic reporting system, Phenics, and a standalone sheet for the MHPSS pilots. MDT meetings started in January 2024. |

### 3.5. Implementation of patient and family centred approach pathway

**3.5.1. Awareness, patient education, and empowerment.** While programme-specific interventions related to the TOC patient and family centred approach pathway, such as patient and family education and empowerment, were not formally prioritised in the first year, in practice, the two pathways operated synergistically. Patient and provider accounts provided evidence that some of these activities and approaches infused the programme's implementation. Education through individual and group awareness sessions was implemented. However, several patient empowerment and education activities outlined in the TOC, such as peer groups to support self-care, and access to paper documents of individual patient record and emergency information for self-management were not implemented in this first phase.

Many of the eight interviewed service users did not perceive receiving awareness or education sessions even though this was provided by social workers at the end of each clinic visit. While ICRC staff felt that these aspects of care were *"underplayed and underemphasised"* by service users, some interviewees noted the importance of being active in improving their NCD control: *When I was working, I had better results, as I was more active (p3).*

Some also suggested that awareness sessions could be conducted while waiting for their consultation, to shorten the clinic visit and limit tiredness. Others requested paper-based leaflets or notes on practical disease management and lifestyle tips. The CAJA and ICRC teams reported that they plan to implement waiting-room based group education sessions on medication adherence, healthy lifestyle, tobacco cessation, and physical activities and additional TOC patient empowerment and education activities but are facing delays due to the current instability in Lebanon. ICRC staff cited several other ongoing health and social care programmes conducted by ICRC and CAJA clinic staff that could generate interest in NCD-related programmes.

*If we start … including older people in our PAVC (people affected by violence) cycle, it will… have a good impact on the NCD programme, because … they go to the community and tell them you know, I am participating in the group (ICRC).*

**3.5.2. Availability and affordability.** Narratives around several intermediate TOC outcomes (availability, affordability of health services, and reciprocal trust) were shared by service users. Some of the most prominent challenges narrated by service users were financial constraints, low-income levels and unemployment. According to service user and staff narratives, their ability to afford medical care, medication, healthy and adequate food, and transport costs had deteriorated significantly during the economic crisis. Availability of medication was prioritised by patients, according to service user and staff narratives:

*The most important (for service users) is the medication. For the patient… (the) doctor, and medications (were service users' main priorities) (ICRC).*

To make integrated NCD care more available and affordable ([Fig 1](ref): TOC map), CAJA and ICRC offered subsidised consultations and medication for vulnerable patient groups:

*I used to buy (my medication) for 10,000 LBP, but it is now for 130,000 LBP. I am unemployed. (Now at CAJA, they say:) "Give us your card". I give them my card. I pay 5,000 LBP, as a small fee for the doctor. (P5)*

*I have access to services and get my medications every month from the centre. Had I been unable to get it from here, I could not have afforded it. Half of the people in Akkar cannot afford medical care. The situation is bad (P3).*

Transport costs limited attendance at weekly MPHSS sessions and ICRC subsidised travel to these for some patients who were considered high-risk. While in-house laboratory testing was planned as part of the model of care, it was determined that this was more expensive for patients, and thus was outsourced to local laboratories:

*We launched the laboratory, but it turned out that if we worked with the main large laboratories, it would still be more affordable for the patient, so we resorted to the second solution (CAJA).*

The ICRC health and PHC teams had planned to expand the "integration" of holistic NCD care, and collaborate inter-sectorally with their economic development division (EcoSec) to implement physical and income generating activities, but this was delayed by the financial and recruitment constraints:

*We would have linked to and started to link with the economic security programmes with gardening projects to have the same target population benefiting from it (ICRC).*

**3.5.3. Patient centred care leading to reciprocal trust building.** While specific person-centred care activities named in the TOC were not emphasised in the first year of implementation, one CAJA staff member reported that their approach to patient care had become more holistic due to the training they received. They described being more aware of the interplay between psychological and physical symptoms, using chest pain as an example of how they now understood that this could be caused by anxiety and depression rather than heart or lung pathology alone. They also described taking these learnings and applying them at other facilities where they worked. Most service users consistently remarked about having high regard and respect for the doctor, nurses, and staff at the CAJA clinic and feeling they were treated with respect.

One interviewed service who lives with Type 1 Diabetes (which is usually managed at secondary care level in Lebanon), was appropriately referred to an endocrinologist. However, she could not adhere to the endocrinologist's advice as she could not afford to buy expensive glucometer test strips. It is notable that she then stopped attending the specialist and returned to CAJA, perhaps because of a more holistic approach, including the availability of glucometer/HbA1c testing and other relevant services, such as MPHSS, and the trust she had built with the team:

*The endocrinologist… asked me to do the tests three times a day for a week... I have the device at home, but I could not afford the strips. I stopped visiting her clinic, and I came back to Doctor (PHC at CAJA) (P1).*

The TOC process also identified the merits of a patient feedback system. While comprehensive patient satisfaction surveys had not been rolled out at the time of the interviews, all eight interviewed service users unequivocally praised the services and communication received from the CAJA staff, highlighting the comprehensive approach to care and the relational effectiveness of their interactions and in-clinic engagement. Monthly satisfaction surveys have been implemented since March 2024, according to ICRC staff.

**3.5.4. Additional unmet needs: access, affordability and availability of non-target medicines, self-care, diagnostic and assistive technology resources.** Effective management of NCDs also requires self-care resources beyond affordable medicines and onward referral for key specialist investigation or intervention. Interviewed service users also wanted wider services to address their co-morbidities, which were not available at PHC level. In PRP, service user and staff narratives discussed supply challenges for prosthetics and shoes in 2023, which was likely due to financial challenges.

### 3.6. Implementation: experience of navigating new processes & co-designing adaptations

**3.6.1. Getting used to new work to co-develop processes.** In the beginning, the implementation narratives were filled with themes of hesitation, and resistance to an increased workload:

*Honestly, in general, when an employee is told that he has more work, he starts to complain at first… He thought it would be a difficult task. (But then,) the process is easy, and it is well carried out (CAJA)*

The magnitude of change and effort was seen to affect productivity, especially when time and resources were limited. The normalisation process of the new practices and ways of working took time, patience, and iterative refinement based on inputs from those involved in executing and operationalising those changes. Implementation was facilitated by clear definitions of roles ("*you know your job; you are responsible for this and this*" – CAJA) which was perceived by staff to minimise conflict and promoted fair distribution of the workload. ICRC's frequent positive follow up supportive supervision and training and collaborative approach to developing adaptations also facilitated buy in from CAJA staff and supported implementation:

*When I noticed (ICRC's) follow-up … you were telling us how to work, how to perform the diagnosis... Such details have developed us more. The main reason that helped us to implement this project is you (CAJA).*

Fig 4 depicts the evolution of one staff member's process of learning, adapting, becoming adept and then proactively contributing to the overall mission of the team.

**3.6.2. Using data systems for real-time learning, reinforcing work processes, and co-developing adaptions.** The revised data collection tools, regular data entry and analysis allowed for immediate quality control within the CAJA team, for example, alerting the GP to missed referral opportunities (Fig 4), and gave the whole team oversight of patients' care, facilitating interdisciplinary shared case management:

*(The) data sheet allows the various teams to see what the GP recommended, their overall clinical outcomes and circumstance, (including) decision making… (regarding) other areas of care; Before, no one else has the information now everyone has the information, … (ICRC)*

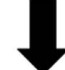
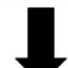
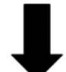

| Overcoming initial unfamiliarity of concept with time & practice | To be honest, the first two or three months, I was confused, and there were still some details that I did not understand. Maybe I thought the issue was more difficult, but when we met more than once and actually started implementing, it was easy. It was such a nice experience when they started with us. I was honestly happy that they were starting with us |
| Acknowledging challenges, & addressing bottlenecks | When we started, data entry was quite difficult for me, in terms of how to collect and receive information from the doctor, the nurse room, and other colleagues. Then, we worked on a different document and made some modifications to the follow-up, which made it easier for me. The doctor checks the information I need, and then I collect and enter it. Before, I had to understand his handwriting and find out what he wrote. It was truly difficult, honestly. |
| Iteratively refining a process until team is satisfied | In general, what I wanted, we have done it. I mean, I wanted the doctor not to write in his own handwriting, and now we have a checklist. I also wanted to find a mechanism to obtain information, whether it was from physiotherapy or psychological treatment, and we also solved this issue; they know what they should do, what data they should give me, what information they should take from patients, and when I should take it from them. These issues can improve. If we notice that the patient needs something more than this, for example, an eye examination, we may be helping and supporting him more. |
| Role clarity; proactiveness & confidence to serve beyond job scope as part of a team | Concerning data entry, I collect all the information, so I can know at the end of the process if something in the patient's file is missing and other colleagues did not pay attention to it, and the patient then, for example, must ask the nurse. I may tell the person who is in charge that the patient must receive physiotherapy or psychological treatment. I am not the one making referrals; in general, the doctor is in charge, whether for physiotherapy or psychological treatment. But, for example, if I see a PHQ test with high results, and the doctor did not pay attention to it, I would bring his attention to this matter. For example, we did not pay attention to the psychological issues, and I had nothing to do with it, as [x] used to manage everything. Now I know the details and the needs of the patients, and I can identify who needs treatment and who does not. It is the same thing for physiotherapy. Other than that, my work is the same. |

**Fig 4. A staff case example - navigation of change and complexity to becoming adept and co-adapting a new practice.**

The ICRC supervisory team (LAA) noted that the diabetic foot assessment was not always conducted in full alignment with the standardized guidelines, particularly in terms of the timing. In collaboration with the CAJA GP, it was identified that a revised appointment structure and refresher session could improve consistency in this regard. The ICRC and PRP teams facilitated this process to ensure more efficient and timely assessments.

**3.6.3. Enabling practices: Flexibility of staff and processes, team-oriented dynamics, and opportunities for problem-solving.** Overall, having flexible staff and processes, and creating team-oriented dynamics and opportunities for problem solving was discussed as enabling practices that allowed adaptations to be ideated and implemented organically:

*An important thing is to learn the dynamics of the place you're gonna work. Because even if you're doing this kind of project, it's still the people there that are doing it. So, you need to really have a good team dynamic… so, basically to understand the team, the local team's dynamics, how they work. (ICRC)*

**3.6.4. Leveraging positionality as a researcher-implementer in pushing implementation and adaptation forward.** Running the research in parallel to project implementation, with the same key ICRC staff members, was seen as a positive factor in pushing implementation and adaptations forward. These dynamics came from having the authority and ability to get team consensus to jointly address a challenge or limitation. For instance, LAA conducted several service user interviews and briefed CAJA staff about their lack of awareness about receiving education, invited the staff's ideas to address the challenge. The LSHTM research team also shared examples of such group NCD education sessions from other organisations and settings.

*I collaborated with the CAJA team to implement group sessions in the waiting room…after discussions with the social worker, who proposed organizing sessions for 3 to 4 patients with similar conditions during waiting times. I was surprised with (her) enthusiasm… she committed to initiating these sessions …. dedicating one day per week for this purpose. (LAA's reflections)*

**3.7. Reach: user uptake of services and ability to receive resources from integrated care**

**3.7.1. Reasons for choice of CAJA clinic: availability, affordability, trust.** The interviewed service users indicated that they heard about the CAJA centre through word of mouth, mostly from family members, neighbours, and community acquaintances. The main reason for choosing to attend this clinic regularly was the affordable medicines and services available, alongside the perceived positive interactions from clinic staff, and especially the doctor and nurses:

In some cases, service users started coming to CAJA because their prior clinics or doctors were no longer available (doctor died during COVID, moved away due to crisis, shut down or no longer serving NCD patients, too expensive due to crisis). Some of them travelled up to 30–40 minutes by taxi, bus or with relatives in a car to attend the CAJA facility.

**3.7.2. Perceptions and uptake of MHPSS: lack of confidence and trust and the need for awareness building.** Narratives from service users, CAJA and ICRC staff suggested a spectrum of opinion around MHPSS in the community. Of note, some interviewed service users were purposively selected due to their non-acceptance of MHPSS services. Two service users declined, two were currently attending, while one was awaiting MHPSS services. Only one of the eight interviewed had attended a course of eight MHPSS sessions. Interviewees expressed psychological distress (*I am psychologically tired*), negative perceptions of MHPSS services, including that they could not solve their life problems, and stigma around mental health needs and medication. Several expressed preferring reliance on faith and religion for support or self-reliance, while others appeared open to talk therapy:

*I might accept attending a session…However, I am afraid of taking a medication because whoever takes it will get used to it and become mad. Now I am facing a difficult situation… I could get over it in one week, or one or two months, and hopefully, then, I will get back to my normal life. (P6)*

In some cases, the study interview provided an opportunity for ICRC staff to address service users' fears and promote referrals to MHPSS and PRP. During patient interviews, LAA took the opportunity to address these concerns and encourage attendance at MHPSS and PRP sessions. In response to patients' reluctance to attend MHPSS, ICRC staff suggested:

*We will not change the reality, but at least there will be a space for you to vent about it and …to learn some new coping strategies to deal with this reality. (ICRC)*

The one service user that completed eight MHPSS sessions learned coping skills and derived therapeutic benefits from speaking about her life concerns in a safe and constructive way.

While many of the interview cohort resisted MHPSS services, the clinic medical professionals and leadership felt that there was an overall shift in community perceptions, where uptake and acceptability were increasingly normalised.

*When we started with the ICRC … I told them that our environment might not be interested in (mental health). Even in people's culture, mental health is a taboo; and if someone suffers … they try to hide it…. That exists in our culture…but with time, experience, and approaches, community has accepted it…. (CAJA)*

### 3.8. Effectiveness: staff and user perceptions around integrated process

**3.8.1. Service user experiences and reactions to new integrated clinic process.** The value and effectiveness of the operational changes executed were noted mainly by ICRC and clinic staff. Even longer-term service users did not perceive many changes in their care or process of care, despite receiving them, like screening for diabetic foot. As discussed, service user narratives mainly centred around being able to receive affordable care that was satisfactory, "comfortable", and more comprehensive than elsewhere. They valued being treated with "respect", feeling cared for, and given time:

*They prepared for me a file…and they immediately provided me with my medications … They did an HbA1C and a urine test. (The nurse) examined my blood sugar level, and my blood pressure and did an electrocardiogram…I only had an examination for my legs/ feet here; I did not have anywhere else. No one has examined the same way they do here, which means that they care more about the patient" (P1).*

However, some also reported an improvement in their diet following dietary advice, with a resultant improvement in their clinical parameters:

*Since I have cholesterol and triglycerides, he tells me to stop eating white bread completely. We apply his recommendations, and we are seeing good results. [I now eat] home meals, beans, and vegetables. I am having better results (P3).*

One unexpected effect of the change in clinic process included tearful reactions to the newly introduced PHQ-2 screening questions. CAJA staff responded to this by drawing the person aside into a private space to support them and flagging their reaction for further exploration by the GP. One interviewed service user affected in this way explained that she was reminded of her circumstances, and it made her emotional (P2).

**3.8.2. Staff perceptions and observations on the effectiveness of the implemented programme for patient outcomes.** Staff narratives described perceived improvements in patients' clinical outcomes and reported symptoms:

*You can see their scores on the (MHPSS) scales pre- and post-intervention, and you can see that there is always an improvement in terms of symptomatology (ICRC)*

They also corroborated patients' positive perceptions, and the value patients placed on the person-centred and caring approach and the service quality:

*Not in my opinion…but in the opinion of patients themselves. This is a distinguished medical service based on European standards, or at least, unavailable standards in a remote area, like Akkar, which they always call the deprived remote area. We are not used to such standards (CAJA)*

This contributed to building trust and resulted in many patients bringing their families to the clinic, aligning with the TOC proposed output of empowering patients and families to engage with services.

*Fit and value of interventions in achieving integration goals.*

ICRC and CAJA team narratives centred around the perceived benefits of the training, streamlined work processes and focus on a holistic approach to care in changing the mindset of staff at the CAJA facility:

*They didn't have this (holistic care) mindset before…You cannot focus only on the disease, because if (patient) is not feeling well, if he's depressed, he's not gonna be better. Even if he took the medication, he's not gonna improve very much if we didn't care for his foot, he might end up with amputation. (ICRC)*

Introducing targeted screening and prevention activities was seen as money well spent by ICRC, who traditionally do not engage in screening activities as it is not part of their remit:

*When you spend money on the preventive approach on the long run, you are saving a lot of money and I'm talking money because when you're talking also patient quality life and everything (ICRC).*

However, it was also noted that more comprehensive, "quality" holistic care required more time, increased workload and that demand had to equal supply, especially in terms of capacity for cross-referral within the facility:

*It is important to take into consideration the patient load and the services… to keep a balance between how many patients we are seeing, the GP is spending how much time with this patient, how is the capacity of the physiotherapist for the referrals (ICRC).*

### 3.9. Broad tacit knowledge and implementation lessons from staff

The staff narratives included useful reflections and experience on implementing the programme, that can serve future replication of similar programmes. These themes mainly anchor on team knowledge, opinions, decision-making and co-development of change and processes. The ICRC team identified several key implementation lessons, which are summarised in Box 1. These include getting to know team dynamics early in the process, and aligning with national health care policy and health system structure:

*Then you must see (what tasks each health care cadre) is allowed to do … because also this plays a big role. It is a big shift in many (countries) in the Middle East…from secondary care to primary care… the decentralization (ICRC)*

Engaging the implementing team early on, studying and developing team dynamics make or break the pro-gramme, according to ICRC accounts. They encouraged iteratively engaging inputs from staff on the ground who will

implement and sustain the integration process, and investing time and resources in team building, role clarity and work expectations:

*It is (important) that people go together to some training, to remove any hierarchy between, you know, a doctor or nurse, physiotherapists and MHPSS, especially so that, in the training, they are all together at the same level; Then also with practical exercises (for) team building, but also realizing that (without) working together, you will not address the needs of the patients… it's really to clarify roles and responsibilities and … to identify a task manager. (ICRC)*

ICRC and CAJA team narratives underlined the importance of ICRC's regular support, supervision and ongoing training, which the CAJA team perceived as inspiring and reinforcing.

### 3.10. Maintenance and sustainability

Resource and financial limitations were prominent themes in anticipating the future of CAJA integrated NCD care programme. One of the assumptions identified during the TOC process was that 'once established, support and budget will be maintained to sustain programme beyond initial implementation phase [13]. However, the programme's sustainability was in serious doubt, according to the CAJA team:

*In general, the local associations and communities… in our regions have no sufficient means for sustainability. Without the support of foreign organizations, no one would have continued…after 2024, if we do not receive new kinds of support, we … will not be able to provide services. (CAJA)*

ICRC staff perceived that the CAJA team's change in mindset and approach would be sustained and that the MOPH also gained valuable learning opportunities from the CAJA integrated care experience. Several contingency plans or factors were in place that may support sustained access to financial, human and medical resources. Firstly, the alignment of PHC and MHPSS elements with national programmes and plans became opportunities for handover to MOPH:

*The National Mental health programme will ensure that it's not only a pilot project anymore, it's going to be a sustainable thing for all PHC's (ICRC).*

Secondly, the ICRC was linking CAJA with a local university to create a pool of volunteer trainee psychologists who would be jointly supervised by ICRC and clinical psychology professors to support the psychology team HR needs.

Ultimately, the programme's sustainability was largely influenced by upstream-level factors: limited and dwindling domestic and international funding for NCDs and PHCs in resource-scarce and humanitarian settings, and ICRCs change in policy regarding their NCD activity:

*So, best case scenario is that … the physical rehabilitation part stays and is taken over by MOPH because this is quite rare and hope that intervention continues. Worst case scenario, it is that… we could just do emergency care and no … regular follow up of the patients. This is neither in (our) hands …nor… the MOPH (ICRC).*

### 3.11. A summative process of early implementation experience

A key theme consistently observed early on and repeatedly during the research and implementation process, was the: '*staff-related human values, motivation, and work ethic as drivers of action, including amidst personal and professional adversity*'.

*it's people they are committed, and we committed to do it and you see how strong also (research team member) is and wants to have everything done like it was planned. So, …when we engage, we go to the end as much as we can (ICRC)*

The ICRC team repeatedly highlighted how the work ethic, motivation to serve and "loyalty" of the CAJA team to their communities were a positive enabler in the change management process:

*The staff of CAJA very welcoming to new ideas, although quite overworked, they… rarely … they're really doing it on the basis that they are happy learning… feeling that … they are of use to their communities (ICRC).*

Positive feedback from patients reportedly further motivated the CAJA team to continue the multidisciplinary team-based approach to care. These broad themes and phenomena around staff related 'power' and empowerment were reiterated in conversations about advice for future replication in other sites.

Fig 5 is a data-derived process generated through the grounded theory analysis process, building on themes and axial codes from emergent data.

## 4. Discussion

Overall, implementing integrated NCD care in this PHC site was found to be feasible, well accepted and valued by participants. The stated programme goals of improved quality of person-centred NCD care, patient empowerment and clinical and quality of life outcomes were met, at least partially. As expected, the theory around how to do it, developed during the TOC process, was challenging under the dynamic conditions in the protracted humanitarian setting of Lebanon. Of the two key pathways to integration identified by the TOC, the multidisciplinary team approach was successfully underway, and, while the patient and family-centred approach was partially implemented, some of its intended outcomes

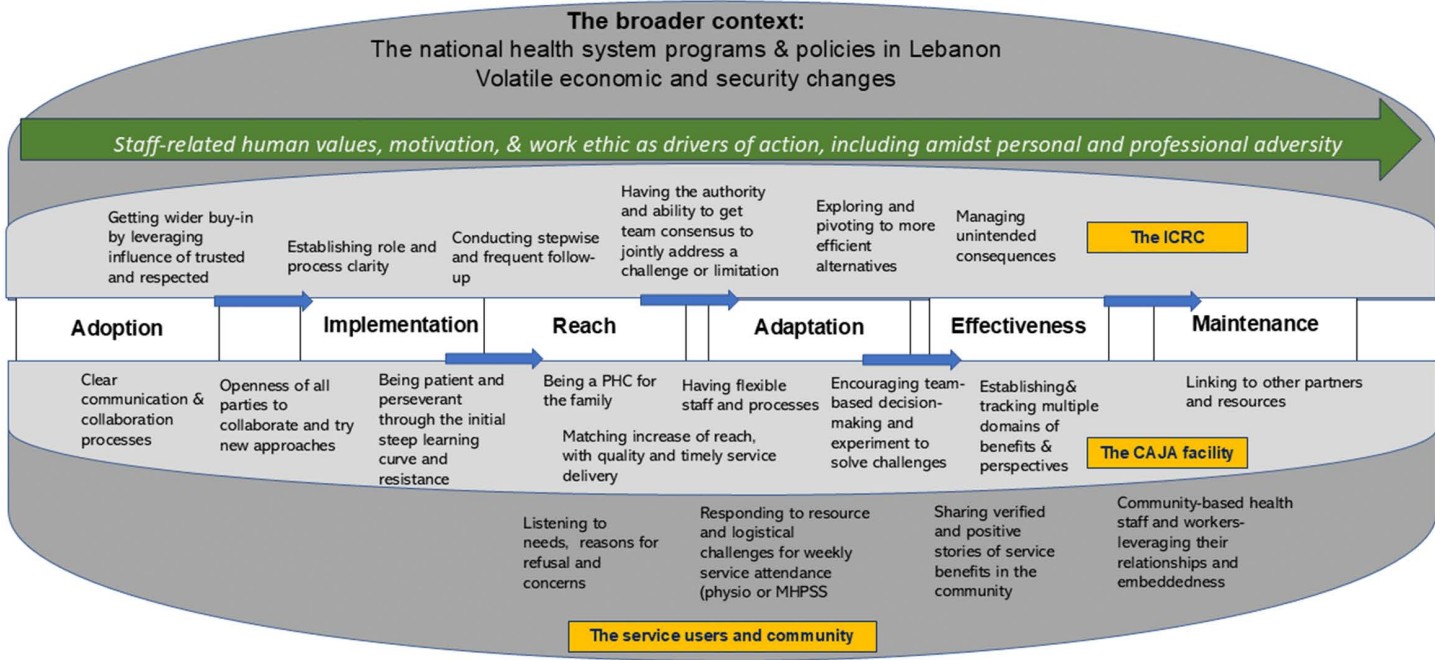

**Fig 5. A summative process of early implementation.**

were achieved since the two pathways overlapped and operated synergistically. Some key implementation lessons were learned by the team, which may be useful for future implementation or scale-up of integrated primary level NCD care in Lebanon and beyond (Box 1).

---

Box 1. 10 key implementation lessons learned to support integrated NCD primary care.

1. Engagement with local health care leaders and government and alignment with local practice and policy is essential.

2. Early engagement and trust-building with PHC staff is essential to build by-in.

3. Study and understand baseline team dynamics.

4. Introduced change gradually with iterative team engagement.

5. Clarify roles and responsibilities and identify a task manager.

6. Invest time and resources in team building; flatten hierarchy, including through joint trainings and role play.

7. Co-develop a streamlined joint information sharing system with the team and regularly review data to support quality of care and service delivery improvements.

8. Hold regular team meetings to reflect on service delivery and available resources, including in the community, existing gaps and needs and individual cases.

9. Empower local staff and co-develop adaptations with them and with service users.

10. Offer regular support, supervision and training.

---

The ICRC and CAJA teams faced multiple external and internal contextual challenges to implementation, including the Lebanese economic crisis and resultant exodus of healthcare workers, and ICRC's internal funding challenges and reorientation back to their founding emergency response remit. The ability of the implementing team to adapt, rather than prioritizing 'fidelity' to the original TOC process derived plan when faced with challenges or opportunities, was a crucial factor in the implementation process. For non-medical cadres, flattened hierarchy was seen to provide psychological security and a safe space to speak up in sharing their challenges and ideas for adaptations. Indeed, the overarching driver of implementation appeared to be the motivation, willingness, quick and responsive problem-solving capabilities, and work ethic of both CAJA and ICRC staff.

The importance of integrating MHPSS with NCD care in humanitarian crises is well recognised, given these conditions frequently co-occur, especially in crisis contexts, and act synergistically. Yet, there are few implementation examples in the literature [43–45]. There was evidence that the training received by staff sensitised them to a more biopsychosocial approach to NCD care. For example, MPHSS and PRP approaches were integrated into the medical consultation as well as within the broader service, with group education sessions including MPHSS, diabetic foot and NCD healthy living topics, and cross referral to the relevant expert teams. Uptake of MHPSS services in CAJA appeared to be affected by stigma and negative perceptions surrounding mental health conditions, treatment and service use, which is consistent with previous experiences at PHC level in Lebanon and the wider region [43,45]. Indeed, structural level interventions to address stigma are already on the MOPH's policy agenda [45]. The service user narratives around limited uptake of MHPSS services, for instance, the belief that they 'will not solve their life problems', may inform messaging around the specific benefits of MHPSS to support utilization of services. Interviewed leaders and doctors welcomed the change in the community's view of MHPSS and may play an important role in promoting uptake, given their positions as respected figures in their communities [45]. We encountered emotional reactions when administering the PHQ-2 and have not

encountered this phenomenon in literature. This may be an important area for future research and action in crisis contexts.

Integration with PRP services at a PHC level for NCD care is a unique facet of this programme. In LMICs, disability prevalence and service use data are sparse. The limited PRP services are mostly found at secondary or tertiary levels of care [46–48]. Although integrated rehabilitation in PHC optimises outcomes [49], what is available tends to be limited in scale, confined to specific types of impairments and predominantly provided by NGOs [47]. Limited LMIC studies suggest that integration of PRP at PHC level is feasible, and, when provided early and consistently, PRP may mitigate and prevent disability and functional loss [46]. In this programme, the service focused on diabetic foot, as this is an area of ICRC expertise that has grown from their experience with traumatic amputation rehabilitation services [50]. Further research in a humanitarian setting on the value and potential of integrated PRP services at PHC level, for example expanding them to cardiopulmonary rehabilitation, may support more health systems to pilot and scale such services.

Integration of data systems across all CAJA PHC services and the transition to a singular digital database was not completely achieved during this initial implementation phase. Collecting data from multiple services in a single patient file allowed the CAJA team to identify missed opportunities for referral, while the ICRC Beirut team's data review and feedback identified the need for booster training and streamlining of processes. Nevertheless, there was a missed opportunity to integrate the Excel with the MOPH driven PHENICS reporting system and the pilot MHPSS datasheet, to reduce replication of data entry work. The challenge of paper-based documentation and the lack of integrated data systems for NCD care in Lebanon have been previously noted and the country is building on the PHENICS system and working with the DHIS-2 platform [15,51]. Ultimately, early and continued regular engagement with MOPH (as outlined in the TOC) is crucial, to support health system strengthening and align with MOPH approach, policy, and systems.

For the second TOC pathway, the patient and family centred approach, several intended intermediate outcomes were achieved despite some of the proposed activities not yet being implemented. These outcomes included achieving awareness, availability and affordability of some health services and resources, and building reciprocal trust in the PHC staff. Data suggest that there was increased awareness and positive feedback about the service among interviewees and within the wider community. Interviewed patients clearly valued consistent access to affordable medication, consultation and laboratory testing and the single interviewed patient receiving MHPSS services had a positive experience. Our findings are broadly consistent with international studies suggesting that integrated care for NCDs can improve user satisfaction, and links them to affordable, acceptable and available necessary services that may serve to manage their overall health and wellbeing [23,27–29]. People with specialised NCD needs, e.g., T1DM or co-morbidities were not fully catered for, as this fell outside of ICRC's remit.

The more downstream patient empowerment and engagement outcomes and activities were partially or not yet achieved in this early timeframe (S1 File). These included building relationships between the facility and surrounding community, enabling self-care through provision of a patient held clinical record and emergency numbers and peer support groups. Reasons for this related to budget constraints, for example, delaying the production of educational or information materials, and ICRC's human resource turnover, which interrupted implementation.

ICRC and CAJA could draw on existing examples when developing these programme aspects. For example, Médecins sans Frontières provide a patient held record for NCD patients, and in Lebanon, a mobile phone-based refugee health records programme (Sijilli) has been developed [2,52–54]. Using mobile phone technology is contingent on levels of smartphone ownership and usage among community members, and may exclude elderly, digital illiterate, women, and economically disadvantaged groups. In Akkar, for example, only 43% of non-Lebanese have smartphones, as opposed to 71% of Lebanese [55]. Peer support groups for PLWNCDs hold promise and have been implemented and undergoing assessment by the Lebanese Red Cross and partners [56,57]. Future empowerment, engagement, and educational activities could also leverage the direct involvement of existing service users, for example, training them as peer champions or

influencers to promote uptake of MHPSS. Cultivating such dynamics aligns with global calls for community engagement and meaningful engagement with PLWNCDs in health programme, policy and research [58].

While there was a sense that service users did not perceive any changes in process or 'integration,' they perceived that they were getting comprehensive care, delivered by respectful and caring local staff, which was of higher quality than that available elsewhere. There is an established national NCD prevention and control plan in Lebanon (NCD-PCP 2016), though several of the early targets such as the establishment of an NCD taskforce, a specific NCD budget and cultivation of a qualified workforce have not been achieved [12]. The inclusion and management of NCDs at primary care levels is already on the MOPH's policy and agenda. Since 2012, the MOPH has focussed on integrating NCDs into a package of care at accredited PHCs [11]. The MOPH has commendably recently introduced a patient complaint system for the national PHC network. Most complaints received related to delays in access to medicines, and referral processes, and poor relational experience and staff attitudes [59]. Fostering an MDT approach to care within the national PHC system, with involvement and task sharing to nursing and social work team members, which may require changes in policy and training, may address some of the workforce issues, support multidisciplinary care and improve patients' experiences [13,60].

There were indications that patients' understanding of their conditions, and knowledge regarding healthy living behaviour improved but, understandably, their ability to engage in these behaviours was constrained by limited resources and the resultant trade-offs and rationing, as previously described in Lebanon, Jordan and elsewhere [14,60–63]. Our previous work demonstrated the huge burden of care experienced by PLWNCDs and their families in Lebanon, and the integrated NCD care model aimed to reduce this burden [14]. The consistent availability of accessible and affordable care and medication likely reduced this burden but not all patient needs were being met. For instance, our findings regarding the needs around non target multimorbidities, the less common type-1 diabetes, and the lack of affordable, accessible and effective referral pathways to relevant hospital-based care are consistent with research in LMIC and humanitarian settings [14,61,64]. Indeed, the illness burden goes beyond health and clinical domains. Trade-offs are made by patients and families in balancing high out of pocket payments including for travel costs, lost productivity and income generation, food, and medicines insecurity. More people-centred and need-based integrated NCD care should not only provide timely, continuous and integrated healthcare across all levels of the healthcare system, but should empower and equip patients and families to self-manage and cope well with their conditions, and live with a sense of autonomy and independence [13,64]. Looking creatively beyond healthcare is key, for example, linking PLWNCDs to income generating activities, such as growing food sustainably. More comprehensive lived experience research may highlight the unmet needs of less visible patient groups and provide context-relevant insights to shape policy and programmes for more integrated NCD care and broader health system strengthening.

Finally, we find that resource and financing challenges continue due to factors beyond ICRC and CAJA's control or influence. The responsiveness and resilience of health systems is weakened over prolonged periods of "polycrisis" [65]. In humanitarian and fragile settings, external or international financing is integral to the delivery of health services and financial decisions are often at the discretion of public and private international actors, and channelled through key international actors, such as the ICRC. International NGOs often operate under short-term funding cycles and strict mandates around lifesaving emergency care and reduction of acute suffering, which may contradict the needs of long-term continuity of NCD care [2,23,66]. While humanitarians are now focusing more on health system strengthening in their NCD programmes, they generally need to exit once capacity is built, or when organizational strategy dictates. Although practical examples of its implementation are limited, the humanitarian-development-peace building nexus has been proposed as a solution to the financial and implementation barriers [67]. This advocates for alignment between humanitarian, development, and peace building actors, in close collaboration with national ministries of health, working to a joint set of plans, policies and systems. For example, diverse actors can engage in pooled procurement and purchasing to negotiate affordable pricing, and more efficiently use limited resources. For humanitarians, handover strategies are planned from the outset, continuity is preserved, and fragmentation avoided.

### 4.1. Study strengths and limitations

This is the first qualitative study, to our knowledge, to describe the integration of MHPSS and rehabilitation into NCD primary healthcare in a humanitarian setting. The study was co-designed with the programme team and implemented in parallel with the programme itself, increasing the quality, richness and depth of insights generated from the research process. Due to the programmatic and contextual challenges outlined above, the numbers and types of interviews undertaken, particularly with patients, was limited. While the team felt that thematic saturation was reached on key areas, additional interviews with a broader patient sample could have further refined or expanded patient-related themes. Moreover, most patient interviews were conducted relatively early in the implementation process, when participants' exposure to integrated care services was still evolving. Interviews conducted later in the programme may have yielded more detailed and nuanced insights into the impact and experience of integration over time. Finally, this study did not include an economic costing or evaluation component that will inevitably influence the sustainability and scale of the programme.

## 5. Conclusions

This research adds to the limited evidence base on integrated NCD care implementation and outcomes in humanitarian settings. In the current age of "polycrisis" and a growing NCD epidemic, it is likely that more people will be susceptible to premature mortality and disability due to a lack of, or interrupted, NCD care if global health actors are unable to course-correct. Innovative solutions are needed to address this. In Lebanon, despite the challenges faced, implementing affordable, acceptable, and available integrated primary level NCD care, MHPSS and PRP services was achievable, supported by ICRC's comprehensive and ongoing training and supervision. A multidisciplinary team of staff in CAJA was equipped, with a mind-set shift, skills, and knowledge to provide more comprehensive, holistic NCD care to service users. Humanitarian actors' ability to adapt rapidly and collaboratively, and the teams' strong human spirit to serve were driving factors in successful implementation. However, larger, upstream factors related to global health financing, policy, and implementation, need to be optimized for integrated primary level NCD care to be sustainable, responsive, and resilient in the age of "polycrisis".

### Supporting information

**S1 File. Theory of change detailed map and supporting information.**
(DOCX)

**S2 File. Interview topic guides (patients and caregivers).**
(DOCX)

**S3 File. Axial codes and themes reported against REAIM-PRISM implementation research domains.**
(DOCX)

**S4 File. CAJA project activities and timeline.**
(XLSX)

**S1 Checklist. Inclusivity in global research.**
(DOCX)

### Acknowledgments

The authors would like to express their heartfelt thanks to the patients, staff and stakeholders who participated in this study.

## Author contributions

**Conceptualization:** Lavanya Vijayasingham, Carla Zmeter, Aya El Khatib, Pablo Perel, Sigiriya Aebischer Perone, Éimhín M. Ansbro.

**Data curation:** Lavanya Vijayasingham.

**Formal analysis:** Linda Abou-Abbas, Lavanya Vijayasingham, Leah Anku Sanga, Sigiriya Aebischer Perone, Éimhín M. Ansbro.

**Funding acquisition:** Pablo Perel.

**Investigation:** Linda Abou-Abbas, Lavanya Vijayasingham, Carla Zmeter, Aya El Khatib, Grace Abou Nakad, Éimhín M. Ansbro.

**Methodology:** Linda Abou-Abbas, Lavanya Vijayasingham, Pablo Perel, Sigiriya Aebischer Perone, Éimhín M. Ansbro.

**Project administration:** Linda Abou-Abbas, Lavanya Vijayasingham, Carla Zmeter, Aya El Khatib, Leah Anku Sanga.

**Resources:** Sigiriya Aebischer Perone.

**Supervision:** Pablo Perel, Sigiriya Aebischer Perone, Éimhín M. Ansbro.

**Validation:** Randa S. Hamadeh, Sigiriya Aebischer Perone.

**Visualization:** Linda Abou-Abbas.

**Writing – original draft:** Linda Abou-Abbas, Lavanya Vijayasingham.

**Writing – review & editing:** Linda Abou-Abbas, Lavanya Vijayasingham, Carla Zmeter, Aya El Khatib, Grace Abou Nakad, Leah Anku Sanga, Randa S. Hamadeh, Pablo Perel, Sigiriya Aebischer Perone, Éimhín M. Ansbro.

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
