## [Decision Letter · Decision Letter 0]

21 Oct 2025

PGPH-D-25-02036

Integrating noncommunicable disease care in a public primary health care facility in North Lebanon: a qualitative study of implementation in a humanitarian crisis

Dear Dr. Ansbro,

Thank you for submitting your manuscript to PLOS Global Public Health. After careful consideration, we feel that it has merit but does not fully meet PLOS Global Public Health’s publication criteria as it currently stands. Therefore, we invite you to submit a revised version of the manuscript that addresses the points raised during the review process.

We look forward to receiving your revised manuscript.

Kind regards,

Dr Buna Bhandari

Academic Editor

Journal Requirements:

Please include a complete copy of PLOS’ questionnaire on inclusivity in global research in your revised manuscript. Our policy for research in this area aims to improve transparency in the reporting of research performed outside of researchers’ own country or community. The policy applies to researchers who have travelled to a different country to conduct research, research with Indigenous populations or their lands, and research on cultural artefacts. The questionnaire can also be requested at the journal’s discretion for any other submissions, even if these conditions are not met.  Please find more information on the policy and a link to download a blank copy of the questionnaire here: https://journals.plos.org/globalpublichealth/s/best-practices-in-research-reporting. Please upload a completed version of your questionnaire as Supporting Information when you resubmit your manuscript.

Additional Editor Comments (if provided):

Reviewers' comments:

Reviewer's Responses to Questions

**Comments to the Author**

1. Does this manuscript meet PLOS Global Public Health’s publication criteria?

Reviewer #1: Yes

Reviewer #2: Yes

2. Has the statistical analysis been performed appropriately and rigorously?

Reviewer #1: Yes

Reviewer #2: N/A

3. Have the authors made all data underlying the findings in their manuscript fully available (please refer to the Data Availability Statement at the start of the manuscript PDF file)?

Reviewer #1: Yes

Reviewer #2: Yes

4. Is the manuscript presented in an intelligible fashion and written in standard English?

Reviewer #1: Yes

Reviewer #2: Yes

Reviewer #1: Good overview of interventions that can be done to improve chronic disease prevention and management. Could also discuss treatments and whether there was any decrease in hospitalizations related to chronic disease as well.

Reviewer #2: These notes provide structured feedback based on key domains of qualitative research appraisal (drawing from established qualitative research evaluation frameworks). The comments are intended to guide authors in strengthening methodological clarity, transparency, and reporting consistency.

Study Aim and Design

• Research question clearly stated and suitable for qualitative inquiry: Yes

• Qualitative design appropriate to address the question: Yes

• Rationale for using a qualitative approach: Yes

Sampling and Participants

• Sampling strategy description (e.g., purposive, theoretical):

Line 240–244: Authors report CAJA service users were purposefully sampled with information on the inclusion criteria and how participants were recruited. Consider providing additional information on the specific type of purposive sampling strategy and rationale for selecting this method.

• Participant characteristics and recruitment methods appropriate and well-described: Yes

• Sample size adequate for achieving data saturation:

Line 249: The authors mention that data saturation was achieved, which supports adequacy of the dataset. However, the total number of participants (or interviews) is not reported. Including this information would improve transparency and allow readers to assess the depth and scope of data collection.

Data Collection

• Data collection methods clearly described: Yes

• Setting and context of data collection specified: Yes

• Reflexivity (researcher’s role, assumptions, influence): Yes

Data Analysis

• Analytic approach described and justified (e.g., thematic, grounded theory): Yes

• Coding and theme development steps transparent: Yes

• Findings supported by illustrative quotes or data extracts: Yes

Trustworthiness and Rigor

• Credibility (triangulation, member checking):

The manuscript indicates triangulation was used. Detailing how member checking was implemented would further strengthen credibility.

• Dependability (audit trail, documentation):

Dependability appears supported through clear analytic documentation and mention of an audit trail. Additional description of coding logs or decision trails could enhance this section.

• Reflexivity or confirmability: Yes

• Transferability (contextual detail for generalization): Providing rich description of the study context could assist readers in assessing transferability.

Reporting and Interpretation

• Findings clearly presented and logically organized: Yes

• Findings linked to existing literature or theory: Yes

• Participant voices presented authentically: Yes

Ethical Considerations

• Ethical approval obtained and stated: Yes

• Informed consent described: Yes

• Confidentiality and participant protection addressed:

Line 309: Could authors provide specific information on how de-identification of participant information was ensured?

Overall Assessment

• Paper clearly written and logically structured: Yes

• Conclusions align with data presented: Yes

• Limitations and implications for practice or research discussed: Yes

Overall, this is a well-organized and thoughtfully executed qualitative study. Enhancing reporting around sampling rationale, participant numbers, and the processes underpinning credibility and dependability would further strengthen methodological transparency.

**Do you want your identity to be public for this peer review?** For information about this choice, including consent withdrawal, please see our Privacy Policy

Reviewer #1: No

Reviewer #2: No

---

## [Editor Report · Decision Letter 1]

28 Jan 2026

Integrating noncommunicable disease care in a public primary health care facility in North Lebanon: a qualitative study of implementation in a humanitarian crisis

PGPH-D-25-02036R1

Dear Dr Ansbro,

We are pleased to inform you that your manuscript 'Integrating noncommunicable disease care in a public primary health care facility in North Lebanon: a qualitative study of implementation in a humanitarian crisis' has been provisionally accepted for publication in PLOS Global Public Health.

Best regards,

Dr Buna Bhandari

Academic Editor